# HUMANS VS CHATGPT: UNCOVERING NON-TRIVIAL DISTINCTIONS BY EVALUATING PARALLEL RESPONSES

## ABSTRACT

The advent of ChatGPT and similar Large Language Models has set the world in an uproar as it is able to generate human-like natural language. Due to the high similarity between the human text and ChatGPT text, it begs the question if the two are truly indistinguishable. In this study, the human-generated content is compared to ChatGPT-3.5, ChatGPT-4, and Davinci-3 using the same technical questions as found on StackOverflow and general questions found on Yahoo Answers. We leveraged Roget's thesaurus to uncover thematic similarities and differences between the human corpora and GPT corpora. We performed a chi-square test on Roget's 1034 categories and found a significant difference in the appearance of words for 365 of them. To uncover the differences in the neighborhoods of the word embedding we utilized the MIT Embedding Comparator to distinguish GloVe base vectors with respect to its trained version on human and ChatGPT corpora. Pre-trained BERT and Sentence-BERT were used to measure the semantic similarity in the answers (on the same questions) given by humans and ChatGPT, which came out highly similar. While that might indicate difficulty in distinguishing ChatGPT and human text, the significant differences in the appearance of words suggested a move towards classification using machine learning models. We observed that various machine learning models performed very well. In summary, we discern disparities and parallels that can be attributed to conceptual, contextual, or lexicographic factors. We endeavor to establish connections between each methodology and these respective categories.

## 1 INTRODUCTION

Large Language Models (LLMs) have been propping up ever since OpenAI revealed ChatGPT. ChatGPT-3.5 and its more reasonable version, ChatGPT-4 have caused a major ripple not just in the tech industry, but all industries across the globe. The immense value of an AI capable of accurately and reliably comprehending and generating natural language has made LLMs enticing to organizations and individuals around the globe. To meet this demand, big tech companies have released their own LLMs: Github's Copilot (Chen et al., 2021) and Google's LaMDA - Language Model for Dialogue Applications (Thoppilan et al., 2022).

In the brief time, LLMs such as ChatGPT have been accessible, ChatGPT-generated content has made its way into all aspects of life. It has already begun affecting education (Zhai, 2022) and professional occupation (Felten et al., 2023) where it masquerades as original, human-generated content. ChatGPT's performance is also impressive even in specialized sectors, it has successfully passed the United States Medical Licensing Exam (USMLE) (Kung et al., 2023). OpenAI's own study takes an early look at the impact such LLMs will have (Eloundou et al., 2023) where they found around 80% of the United States workforce would have at least

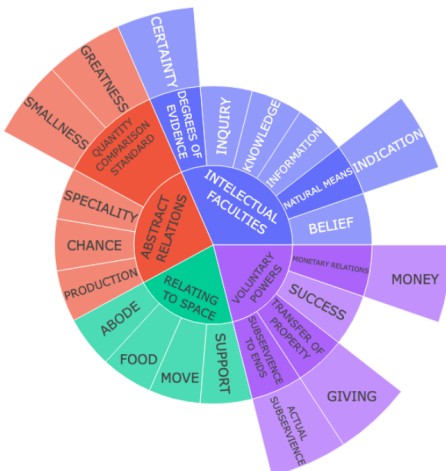

Figure 1: This sunburst presents a view of the most significant Roget's Theme and Categories based on the chi-square test performed on the words appearing per Roget's Category. The top 20 categories with the highest chi-square score and p-value < 0.05 are shown.

10% of their tasks altered due to these LLMs. The study (Eloundou et al., 2023) finds that 19% of workers across different disciplines may have over 50% of their work affected, increasing the urgency for research.

Recent literature has attempted to model machine learning and deep learning models to distinguish human text and text generated by ChatGPT (Guo et al., 2023; Mitrović et al., 2023; Shijaku & Canhasi). Based on their results, it is possible to differentiate the two using machine learning techniques. Roget's Thesaurus, an English Language Thesaurus, written by Peter Mark Roget, a British Lexicographer (Roget & Roget, 1925) can also aid us. It is an excellent resource similar to WordNet (Fellbaum, 2010). It shines in its ability to measure semantic similarity (Jarmasz & Szpakowicz, 2003) and produce word clusters with higher correlation (Jarmasz, 2012) when compared to WordNet. Longer discussion regarding LLMs and methods to classify their text output from humans can be found in Appendix A.

In addition, this paper delves deeper into non-trivial distinctions between using various evaluation metrics. We attempted to analyze syntactic, semantic, and lexicographic patterns between the two. Furthermore, we examined how the same words are used in diverse contexts by humans and ChatGPT by using Roget's thesaurus as seen in Fig 1. In order to make the linguistic comparisons largely fair we collected our own dataset and structured it as a parallel corpus between humans and ChatGPT. The contributions made by this paper are:

- Novel parallel datasets of text data where ChatGPT-3.5, ChatGPT-4 and Davinci-3 are prompted to answer questions and generate video scripts

- We compare the Parts-Of-Speech tag distribution, BLEU score, stopword distribution, and semantic similarity using BERT models between parallel human responses and GPT-generated responses

- Performing comparison of human-generated text to ChatGPT-generated text utilizing Roget's Thesaurus as a human-grounded knowledge base

- Compare lexicographic differences using Machine Learning approaches

## 2 DATASET

### 2.1 DATASET SUMMARY

To accurately compare human-generated text and the text generated by ChatGPT, a novel dataset is constructed. A total of 13,496 data points have been collected. 6250 are human-generated content and the associated question or title from three platforms: StackOverflow, Yahoo Answers, and YouTube. 6144 data points are from GPT models answering the same questions or producing content for the same title. 1106 data points are of ChatGPT being told to pretend to be a user of StackOverflow or a YouTuber when generating content. Examples of how the dataset is structured are shown in Table 1. A summary of the data collection can be found in Table 2. The datasets have been given acronyms for the rest of this paper for brevity. Yahoo Answers by humans is shortened to YA-H and by Davinci-3 is YA-D3. StackOverflow answers by humans is shortened to SO-H, by ChatGPT-3.5 without context is SO-CGPT3, and with context is SO-CGPT3-C. YouTube transcripts, which are scripts read out by humans in videos, is shortened to YT-H and the scripts generated by ChatGPT-4 are YT-CGPT4.

Table 1: Examples of responses given by a human and generated by a GPT model on the same topic

| QUESTION | HUMAN RESPONSE | GPT RESPONSE |
|---|---|---|
| How do I most elegantly express left join with aggregate SQL as LINQ query? | A solution, albeit one that defers handling of the null value to the code, could be:DateTime yesterday = DateTime.No... | To express a left join with an aggregate function as a LINQ query, you can use the GroupJoin method... |
| Death From Space — Gamma-Ray Bursts Explained | Imagine if you could gather the energy from every star within a hundred million light years. From thousands of galaxies... | Hello, space enthusiasts! Welcome back to our channel. Today, we're going to delve into one of the most fascinating and powerful events... |

### 2.2 DATASET CONSTRUCTION

#### 2.2.1 COLLECTING HUMAN RESPONSES

An existing dataset of Yahoo Answers (soumikrakshit) was used in collecting human data. This is advantageous as it ensures that all the content was generated by a person before ChatGPT-generated content appeared in forums. The title and body of the question were also concatenated to produce the "prompt". StackOverflow provides a snapshot of their database, which is exposed for research purposes. We queried it to acquire the titles, questions, and answers in HTML format. The data was cleaned using the BeautifulSoup library's HTML parser. The title acts as the main question and the details of the question are given in the question body, therefore the title was concatenated with the body to create the "prompt" for ChatGPT. The top accepted answer per question was collected as human data. With the recent release of ChatGPT-4 and its ability to generate long-form content, it was necessary to compare that to human-generated content such as YouTube videos. OpenAI's transcription model, Whisper (Radford et al., 2022), was utilized to extract transcriptions of the videos. The "medium" model of Whisper was used with 769M parameters.

Table 2: The number of human-generated and GPT-generated data points collected from the platforms.

| Platform | Human Datapoints | GPT Datapoints | Contextual GPT Datapoints |
|---|---|---|---|
| StackOverflow | 1188 | 1188 | 1000 |
| YouTube Transcripts | 106 | - | 106 |
| Yahoo Answers | 4954 | 4954 | - |
| **Total** | **6250** | **6144** | **1106** |

### 2.2.2 COLLECTING GPT RESPONSES

ChatGPT-3 is OpenAI's chatbot built on top GPT-3.5. In order to collect data from ChatGPT-3, the "prompts" created from the questions from StackOverflow and Yahoo Answers were fed into a new instance of chat. Afterward, the generated answer was collected. For contextual answers to StackOverflow questions, the prompt was modified by adding a phrase before the question "Answer as a StackOverflow user to the question ..." to the user input. ChatGPT-4 is the next iteration of ChatGPT-3. Its headlining features are more reasonable and generates long-form content. Instead of plainly using the title for the prompt, ChatGPT-4 was asked to write a YouTube script with the title. An example of the prompt is, "Write a YouTube script, only the content without scene direction or mentioning the host. Do not write out sections in your box brackets. with the title: …".

## 3 METHODOLOGY

### 3.1 EVALUATION METRICS

### 3.1.1 PARTS-OF-SPEECH, STOP WORDS AND BLEU SCORE

The Parts-Of-Speech (POS) distribution in human-generated text and ChatGPT-generated text were analyzed. Once tagged, the corpora containing human-generated text were compared to the corpora containing ChatGPT-generated text. The pairings for the comparisons are as follows: YT-H and YT-CGPT-4; YA-H and YA-D3; SO-H and SO-CGPT3; SO-CGPT3 and SO-CGPT3-C. These pairings are kept for all experiments. For each pairing, the most frequently occurring POS tags have been compared between human-text and ChatGPT-text. The second area to be looked into for potential differences is the occurrence of stop words. We observe if the usage of stopwords differs between humans and ChatGPT. The Bilingual Evaluation Understudy (BLEU) scores between human-generated text and ChatGPT-generated text have also been calculated. BLEU scores have been evaluated from BLEU-1 to BLEU-4. The number indicates the level of n-gram precision. For example, BLEU-1 represents unigram precision, BLEU-2 represents bigram precision, and so on.

### 3.2 MAPPING TO ROGET'S THESAURUS

As described by Roget, Roget's thesaurus is "... a collection of the words it [the English language] contains and of the idiomatic combinations peculiar to it, arranged, not in alphabetical order as they are in a Dictionary, but according to the ideas which they express" (Roget, 1852). Each of the answer pairs were fed through an algorithm that takes the words in the text and maps them to a corresponding category in Roget's thesaurus. An example of a word being mapped is given in Fig 2. The mapping starts out broad and gets increasingly more specific. More details are included in Appendix A.

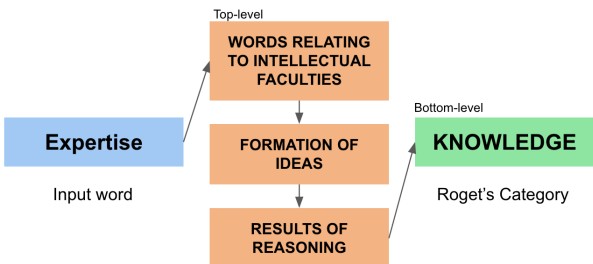

Figure 2: Example of mapping a word into Roget's Thesaurus. The word "Expertise" is related to the category of "Knowledge". Which in turn is under various themes denoted in the diagram.

### 3.3 COMPARING WORD NEIGHBORHOODS

Using Global Vectors for Word Representations (GloVe) (Pennington et al., 2014), we capture the contextual differences of word usage between humans and ChatGPT. We map words used by both humans and ChatGPT into a high-dimensional space and leverage GloVe to cluster all the words. Once the word embeddings are mapped, the Embedding Comparator introduced by MIT (Boggust et al., 2022) is used to perform a global comparison of embedding spaces and analyze local neighborhoods as well. For our experiment, we gather embeddings from three corpora: the GloVe algorithm trained on Wikipedia and Gigaword 5 which we take as the base, GloVe trained on a human corpus, and GloVe trained on a GPT corpus.

### 3.4 HIGH-LEVEL TEXTUAL COMPARISON USING BERT MODELS

For a high-level comparison between human-generated text and ChatGPT-generated text, pre-trained BERT and Sentence-BERT models were deployed. The pre-trained BERT model assessed the same pairings of data, capturing the output and calculating the cosine similarity between human-text embedding $\vec{H}$ and GPT-text embedding $\vec{G}$ across 13 hidden layers. T-SNE plots visualized these embeddings for each layer. Concurrently, Sentence-BERT, specifically the "all-MiniLM-L6-v2" model from Hugging Face, was utilized for evaluating semantic similarity. The pairwise cosine similarity of these embedded data points was calculated and represented through stacked histograms, facilitating a detailed analysis of the similarity between human and GPT corpora.

### 3.5 MODELLING WITH MACHINE LEARNING AND BASELINE BERT

Each data pairing was cleaned through the removal of URLs, punctuation, and stop words. The data were encoded using the Term Frequency-Inverse Document Frequency (TF-IDF) vectorizer. The train test split was 75:25. Various classifiers were trained on a binary classification task using traditional machine-learning models. The models used were Support Vector Machine Classifier (SVM) (Hearst et al., 1998), Naive Bayes classifier (Rish et al., 2001), and eXtreme Gradient Boosting (XGB) (Chen et al., 2015). Afterward, an exhaustive feature reduction was performed on linear SVM to further see how distinct the lexicographic differences between the classes are. The same task was achieved through baseline BERT, with the BERT pre-processor "bert-en-uncased-preprocess" and encoder "bert-en-uncased-L-12-H-768-A-12" from TensorFlow being used. The encoder uses 12 hidden layers (i.e., Transformer blocks), a hidden size of 768, and 12 attention heads with the Adam optimizer is used and binary cross-entropy loss function.

# 4 RESULTS DISCUSSION

## 4.1 HUMAN VS CHATGPT CLASSIFICATION

XGB performed the best on the datasets of SO-H vs SO-CGPT3 and YA-H vs YA-D3 with an accuracy of 92% and 78% respectively. SVM has performed the best in YT-H vs YT-CGPT4 and SO-CGPT3-C vs SO-H with accuracy of 96% and 94% respectively. XGB and SVM have tied at 83% in SO-CGPT3-C vs SO-CGPT-3. NB, while having a lower accuracy score than XGB and SVM, has also performed well in text classification. Our baseline BERT performs similarly. We believe further fine-tuning the parameters could improve the accuracy but our goal was not to optimize for classification. The complete list of results is found in Table 3. Performing feature reduction on linear SVM results in improvements across all data pairings except YouTube and Yahoo Answers as we see in Figure 3. The high performance of the statistical machine learning model lends credence to the idea that there are enough lexicographic differences between human text and GPT text to distinguish the two.

Table 3: Model performance in classification task across datasets

| Dataset | Model | Accuracy | ROC AUC | F1 Score |
|---------|-------|----------|---------|----------|
| SO-CGPT3 | SVC | 90% | 0.90 | 0.90 |
| vs | NB | 86% | 0.86 | 0.86 |
| SO-H | XGB | 92% | 0.93 | 0.93 |
|  | BERT | 77% | 0.79 | 0.77 |
| YA-D3 | SVC | 77% | 0.77 | 0.77 |
| vs | NB | 74% | 0.74 | 0.75 |
| YA-H | XGB | 78% | 0.78 | 0.78 |
|  | BERT | 80% | 0.84 | 0.79 |
| YT-CGPT4 | SVC | 96% | 0.97 | 0.96 |
| vs | NB | 94% | 0.93 | 0.94 |
| YT-H | XGB | 94% | 0.94 | 0.94 |
|  | BERT | 66% | 0.69 | 0.67 |
| SO-CGPT3-C | SVC | 83% | 0.83 | 0.83 |
| vs | NB | 80% | 0.80 | 0.80 |
| SO-CGPT3 | XGB | 83% | 0.83 | 0.83 |
|  | BERT | 79% | 0.86 | 0.80 |
| SO-CGPT3-C | SVC | 94% | 0.94 | 0.94 |
| vs | NB | 90% | 0.90 | 0.90 |
| SO-H | XGB | 90% | 0.90 | 0.90 |
|  | BERT | 75% | 0.83 | 0.75 |

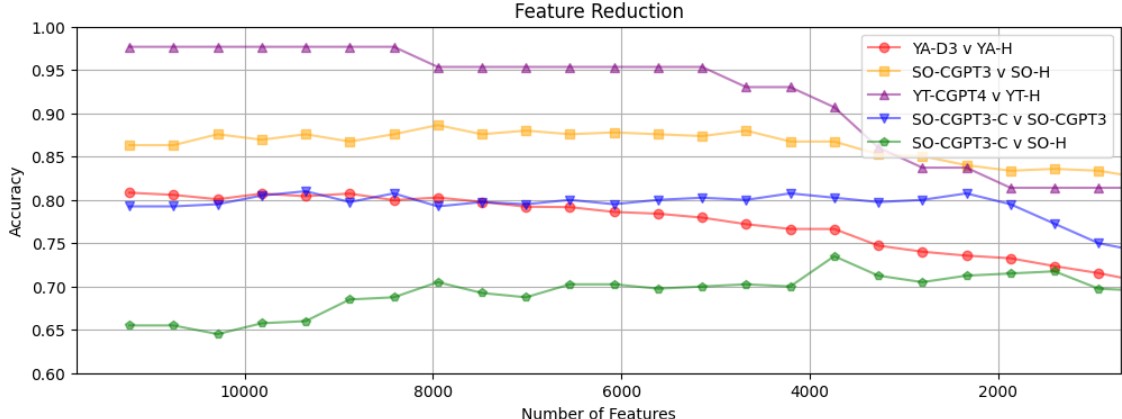

Figure 3: Linear SVM's accuracy tracked across feature reduction steps

## 4.2 FINDINGS FROM ROGET'S CATEGORIES

The mapping of the corpora to the 6 main Roget's Themes has shown little difference between humans and ChatGPT. The high similarity of the themes means human-generated text and ChatGPT-text cannot be separated on a thematic basis when comparing the parent themes. When we compare the base Roget's categories, we find that many of the categories show a strong relationship. The chi-square score is calculated for each of these base categories mapped to both humans and ChatGPT. The p-value is calculated for all of these mappings after calculating the chi-square value. For 365 of them, it is less than 0.05 which means that the observed data is extremely unlikely to have occurred by chance alone under the null hypothesis. This could convey the idea that the same concepts are used in different context by humans and ChatGPT which could be a distinguishing feature. The top 20 most significant Roget's categories and their respective themes are illustrated in Figure 1.

## 4.3 POS TAG DISTRIBUTION, STOP WORDS AND BLEU

The difference in POS-tag distribution for the 4 pairings has been normalized. Illustration can be found in A in Figure 7. The distribution of POS tags is observed to be largely similar between humans and ChatGPT. One difference is, GPT models tend to use more noun singular (NN) when generating qualitative content. When generating technical content, humans tend to use significantly more NN. However, the difference is slight at 0.04. When ChatGPT-3.5 was asked to mimic a human while answering StackOverflow questions, the difference in the distribution of POS tags was minimized. Indicating that if properly prompted, ChatGPT is capable of answering in a more human-like manner in one-shot answers. The results of the stopword analysis were determined to lack statistical significance and as such they have been included in Appendix A. Ultimately, it is evident that POS-tag distribution and stopword analysis cannot be a reliable indicator of discriminating between humans and GPT-generated content.

BLEU score gives an interesting insight. The specific scores can be found in Appendix A under Table 4. BLEU-1 has the highest overall score with the highest being between YA-D3 and YA-H at 0.937. The high BLEU-1 score indicates that the they use similar vocabulary. BLEU-3 and BLEU-4 have very poor scores which indicates sentence structures are possibly different.

## 4.4 BERT AND SENTENCE-BERT REPRESENTATION ANALYSIS

The same pairings of human and GPT data have been propagated through BERT and SBERT. Afterward, the cosine similarity scores have been calculated and normalized. The scores are plotted as a stacked histogram in Figure 4 for SBERT and T-SNE plot in Figure 5 for BERT. YA-D3 vs YA-H have the least in common in terms of semantic textual similarity. It is the only pairing that has a significant portion of its cosine similarity in the negative. The other pairings all have high cosine similarity scores. The highest being between SO-CGPT3-C and SO-H. This is further evidence that discriminating between ChatGPT and humans at a high level is a challenging task. The high-level representations in the pre-trained BERT model appear to be insufficient in discriminating between the pairings. On a side note, the similarity between SO-CGPT3-C and SO-H being so high is evidence that when prompted to mimic a human, the answers produced by ChatGPT mirror closely those given by humans.

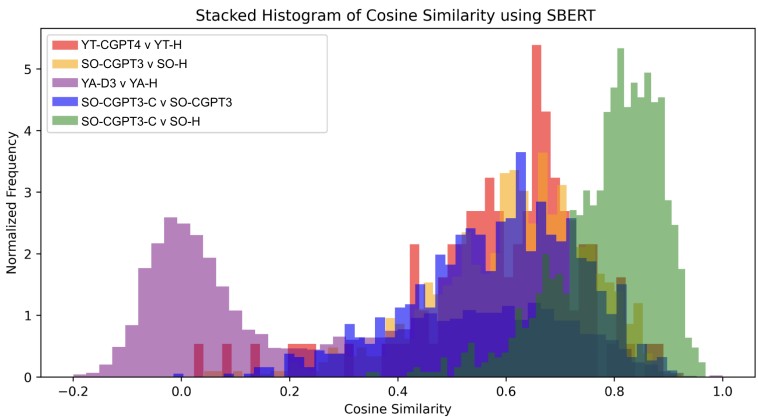

Figure 4: SBERT cosine similarity, illustrating pairwise semantic similarity.

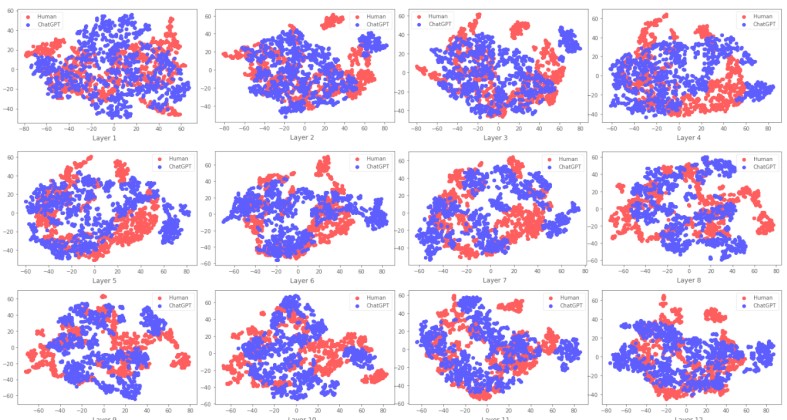

Figure 5: T-SNE plots of layers of pre-trained BERT of StackOverflow embeddings. Red points are SO-H and blue points are SO-CGPT3.

### 4.5 WORD NEIGHBORHOODS

The embeddings from the base and trained GloVe models were obtained and analyzed using the Embedding Comparator. We find that when we compare the common words used by humans and ChatGPT and observe their local word neighborhood, the neighbors are different. An example are the words "comfort" and "terrorism", whose local neighborhoods are illustrated in Figure 6(a) and Figure 6(b) respectively. The neighborhoods in which the word "comfort" is found in a high-dimensional mapping of the human corpus are different than that of Davinci-3's corpus. This is indicative that even if the words used by humans and GPT models are the same, the usage of these words differs in their contexts. Thus concepts they deliver are different.

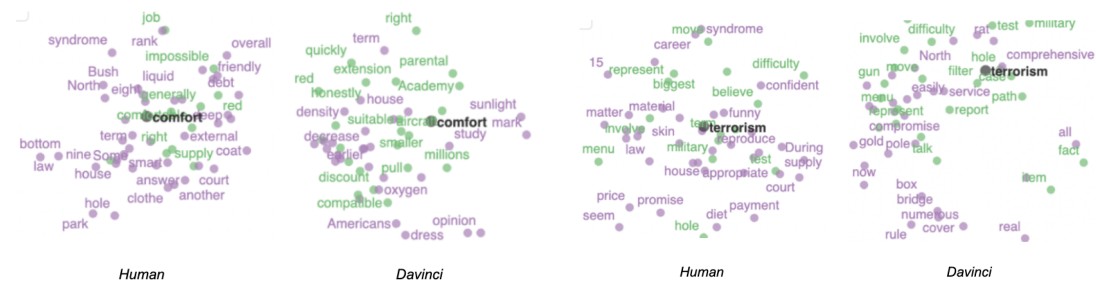

(a) Local neighborhood for the word "comfort"  (b) Local neighborhood for the word "terrorism"

Figure 6: Illustration of word neighbourhoods found using the Embedding Comparator (Boggust et al., 2022) on GloVe embeddings trained on human text (YA-H) and Davinci text (YA-D3).

## 5 LIMITATIONS AND FUTURE WORK

While the results are promising, it is important to note the limitations of this study. The data from the GPT models were analyzed as-is, unlike in real-world scenarios where ChatGPT's output is often manually modified or rephrased using a tool. Future work can include collecting a larger parallel dataset with a larger variety- having ChatGPT answers with and without context. Another would be to perform the same experiments but across different languages to see whether the differences outlined in the paper hold true across various languages.

## 6 CONCLUSION

Classification of human text and ChatGPT-generated text does not require a complex model. An observation made in (Ippolito et al., 2019) is that generation models produce high-likelihood words. This is in contrast with humans who are more likely to introduce statistical anomalies in their texts. Suggesting a significant difference in the vocabulary used by the two. This idea is reinforced by our findings. The analysis of the appearance of Roget's categories reveals that there is a non-random pattern in the way humans use words that differ from ChatGPT. And we observe this trend again when analyzing the word neighborhoods for humans and ChatGPT in embedding space. In both texts, they appear to remain contextually relevant and as such there is high similarity when using BERT or SBERT models. In conclusion, syntactic and contextual differences are insignificant but conceptual differences appear to be significant. Lexicographic differences are also significant and can be picked up easily using machine learning approaches, which explains the high performance.

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

## A    APPENDIX: RELEVANT WORK

### A.1    LARGE LANGUAGE MODELS

Since the advent of ChatGPT and the subsequent popularization of LLMs, there has been a push toward developing more sophisticated models. Simultaneously, methods to distinguish AI-generated text from humans were also being developed. The first popular LLM was Bidirectional Encoder Representations from Transformers also known as BERT (Devlin et al., 2019). It was the first, popular foray into using LLMs to accurately understand natural language and has also been used to generate cohesive sentences (Wang & Cho, 2019).

ChatGPT is a chatbot developed by OpenAI developed first, on top of their third generation of Generative Pre-trained Transformer, GPT-3 (Brown et al., 2020). It is an auto-regressive Large Language Model (LLM) consisting of 175 billion parameters. When it was first released, it excelled at Natural Language Processing (NLP) tasks such as translation and answering questions. Even on its first release, the OpenAI team found

that if they asked human evaluators to distinguish between GPT-3 generated and human-written articles, they found it a difficult task. Davinci-3 is another GPT-3 powered LLM (Zong & Krishnamachari, 2022) that is the closest older relative of ChatGPT. It is capable of performing natural language tasks such as question answering and code generation with high-quality output.

ChatGPT-4 is OpenAI's next version of ChatGPT built on-top of GPT-4, a large-scale, multi-modal model (OpenAI, 2023). It's touted as being capable of human-level performance on academic and professional tasks, better than the previous GPT-3 model. For example, when taking a simulated bar exam, GPT-3.5 scores in the bottom 10% while GPT-4 scores in the top 10%. GPT-4 also comprehends user intention better which allows it to generate better answers when prompted. When prompted with 5,214 questions, the answers generated by GPT-4 were preferred over GPT-3.5 for 70.2% of time.

## A.2 DISTINGUISHING GPT AND HUMAN TEXT

Similar studies have begun trying to distinguish text generated from LLMs from human-generated texts (Guo et al., 2023)(Mitrović et al., 2023). Their experimental setup is one in which they ask ChatGPT to the same content produced by humans and then compare the two. These studies put an emphasis on using deep learning techniques to find distinguishable patterns. However, a few recent literature has shown that traditional machine learning techniques work decently well in classifying text between humans and ChatGPT (Shijaku & Canhasi). An observation many have made is that text generated by GPT models lack statistical abnormalities, which makes it difficult for statistical models to distinguish it (Ippolito et al., 2019). However, this implies human text has higher rates of abnormalities which can be easily picked up by simpler models.

Another important resource that can be used to qualitatively differentiate between human text and GPT text is Roget's Thesaurus. Roget's Thesaurus is an English Language Thesaurus originally written by Peter Mark Roget, a British Lexicographer (Roget & Roget, 1925). It is an excellent resource similar to WordNet (Fellbaum, 2010), a large-scale lexical database. The area in which it shines is its ability in measure semantic similarity (Jarmasz & Szpakowicz, 2003) and produce word clusters with high correlation (Jarmasz, 2012) when compared to WordNet. Which could be utilized to paint any thematic differences between human and GPT text.

BERT has been shown to classify texts similar to human estimates. (Koroteev, 2021) Pretrained-BERT has been used before to propagate the inputs through each layer and then calculating the pairwise cosine similarity to analyze textual similarities. Sentence-BERT (SBERT) (Reimers & Gurevych, 2019) is a modification of BERT which utilizes a siamese and triplet network to also calculate text-similarity in a pairwise manner using cosine similarity. It outperforms BERT in regard to this task.

## A.3 MAPPING TO ROGET'S THESAURUS

Roget's thesaurus can be a powerful tool to identify semantic differences in groups of text. It classifies the ideas expressible by the English language into six classes: Abstract relations, Space, Material World, Intellect, Volition, Sentient, and Moral Powers. There are sections within these six classes, and under these sections, there are almost 1,034 categories.

Initially, the words are mapped to the six main classes. Afterward, they were mapped to Roget's categories which are the words present in the thesaurus that relate most closely to the input words. The use of the thesaurus helps us compare the differences in ideas expressed by ChatGPT and humans.

For each of the categories, the chi-square value was calculated to see if there was a relationship among them. The chi-square is a statistical test that examines the differences between categorical variables from a random sample in order to determine whether the expected and observed results are well-fitting. Using this value and the degrees of freedom, the p-value is calculated. The p-value describes the likeliness that the data would

have occurred under the null hypothesis of the statistical test. Categories with p-value < 0.05 are considered as the result is considered significant. The most significant among these from the YA-H and YA-D3 with the highest chi-square values are plotted in Figure 1. The POS tag illustration is shown in Figure 7

### A.4 STOP WORD ANALYSIS AND POS TAG ILLUSTRATION

We observe the use of stopwords expressed as a percentage of the dataset in 8 (a), and the normalized frequency of the top 10 most frequent stopwords in 8 (b). There is no significant difference when observing the total amount of stopwords used between humans and ChatGPT-generated text. Therefore, stopword usage is not a feature that can be used to discriminate between the two.

The text data has been propagated through the 13 hidden layers found in pre-trained BERT. Figure 5 is a series of T-SNE plots of the CLS Tokens that have been extracted from each hidden layer when SO-H and SO-CGPT3 have been propagated. There is a high overlap between the two.

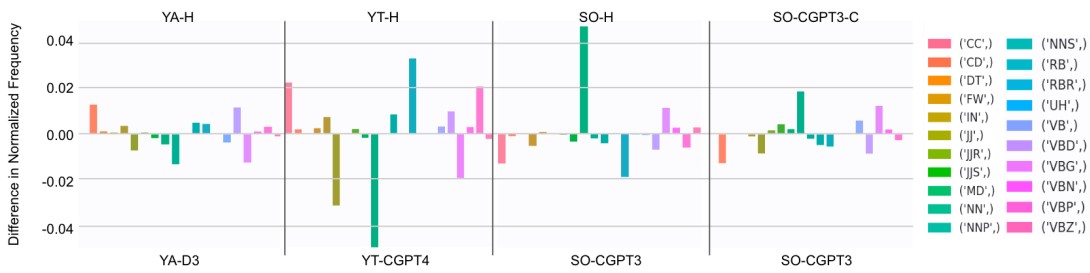

Figure 7: Difference in distribution of POS-tags illustrated with normalized values

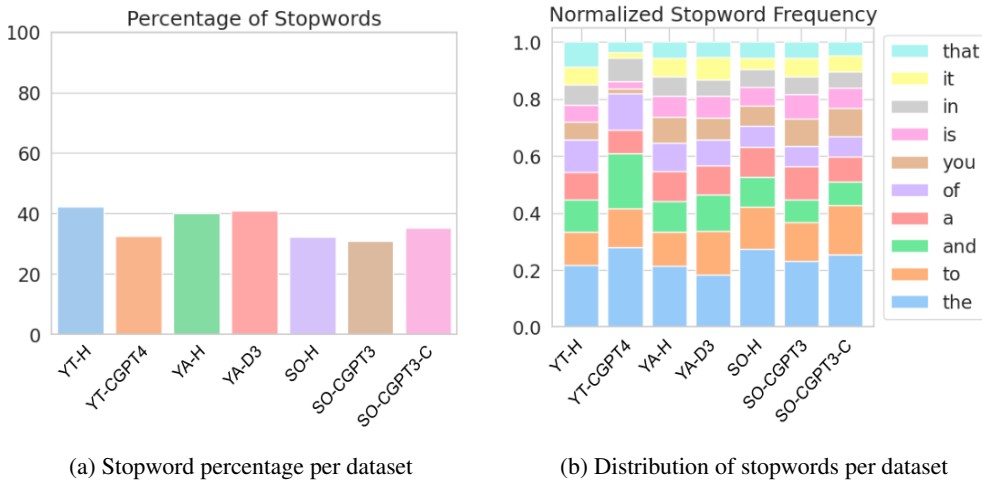

(a) Stopword percentage per dataset
(b) Distribution of stopwords per dataset

Figure 8: Difference in distribution of POS-tags illustrated with normalized values

## A.5 BLEU SCORES

Table 4: Average BLEU Scores per pairing.

| Pairing | BLEU-1 | BLEU-2 | BLEU-3 | BLEU-4 |
|---|---|---|---|---|
| SO-CGPT3 vs SO-H | 0.806 | 0.408 | 0.123 | 0.029 |
| YA-D3 vs YA-H | 0.937 | 0.552 | 0.193 | 0.049 |
| YT-CGPT4 vs YT-H | 0.734 | 0.037 | 0.001 | 0.000 |
| SO-CGPT3-C vs SO-CGPT3 | 0.684 | 0.047 | 0.001 | 0.000 |

