# OpenReview forum: "Humans vs ChatGPT: Uncovering the Non-trivial Distinctions by Evaluating Parallel Responses"
_ICLR.cc/2024/Conference — Submitted to ICLR 2024_

### Official Review · Reviewer_spBk · 2023-10-26

**Soundness:** 1 poor
**Presentation:** 2 fair
**Contribution:** 1 poor
**Rating:** 3
**Confidence:** 5

**Summary:**

The paper describes work on exploring different combinations of linguistic features (which the authors called “non-trivial distinctions”) between human and machine-generated content (ChatGPT). The breadth of analysis the author/s conducted include part-of-speech tag distribution, stop words use, semantic similarity from embedding representations, and theme selection from Roget’s thesaurus. The authors then trained classification models using different models (SVM, Naive Bayes, XGB, BERT) using TF-IDF to which the results do not have extensive ablations with the preliminary features extracted. The author/s motivate the work and results by claiming that these types of studies on analyzing differences between styles of human and machine writing does not require complex models, however the depth of experimentations and contribution has not been met by the paper to produce a cohesive, impactful result.

**Strengths:**

The paper shows an effort toward further understanding some interesting insights into linguistic variables between humans and machines (ChatGPT). The insights produced from the paper may benefit specific communities, particularly looking at stylistic and linguistic differences between content produced by the two groups. The paper definitely has potential and I hope the authors take into consideration the feedback in order to improve the rigor and depth of research required by the conference.

**Weaknesses:**

The overall study seems to be just a mix-and-match of whatever tool is available for the author/s that will be able to give some form of analysis without proposing any novel approach to shed new insights in the growing field of discriminating between machine and human-generated texts. In addition, most of the analysis is weak or unmotivated. (For example, what are non-trivial features? Why is there a need for analysis of stopwords use? In what way does the use of stopwords shed light on human and machine-generated text?) Moreover, there is no comparison of existing works in literature including already established classification systems such as DetectGPT, GPT-Zero, etc with the trained models from the paper. These limitations in the paper severely affect the quality of the work and its inclusion for ICLR.

Another demerit of the paper is the lack of concrete discussion of the data used. The author/s claim that the data is novel but provides very limited information on how it was collected particularly on important details such as language, statistical characteristic (how long, how many words), context of where the comments or questions are about, genre or domain, etc. Moreover, the author/s seem to use the collected data for querying ChatGPT but other prompt-ready QA datasets have already been constructed. Why did the authors not use or combine this instead?

Improvement of how the motivation of the paper is pieced together, especially in the Introduction section, should be strictly addressed to increase readability and in order for readers to appreciate the purpose of the study (see comments on structuring below).

Minor comments on structuring and flow of discussion that need to be addressed:

The introduction section needs to be improved as it does not sound technical/scientific. Vague and overgeneralized claims with no basis should be removed particularly in the first sentence “Large Language Models (LLMs) have been propping up ever since OpenAI revealed ChatGPT”. LLMs have already been an active research even before ChatGPT. The sentence “ChatGPT-generated content has made its way into all aspects of life.” is also vague and too highfaluting. The introduction of ChatGPT had some impact, yes, but you need to ground this technical with figures instead and not with these types of statements.

There is no ChatGPT-4. Do you mean GPT-4? ChatGPT was packaged on its own using the GPT-3.5 Turbo model while GPT-4 is a separate model that was only integrated into the ChatGPT system as another option.

**Questions:**

1. What is non-trivial in the context of the study? And how does it differ from previous works (ex. https://arxiv.org/abs/2306.07799, https://arxiv.org/abs/2204.05185)  doing an evaluation of linguistic differences between humans and generative models (ChatGPT, GPT-2)? The term is only mentioned twice in the paper with no further discussion at all which makes any reader fail to appreciate the overall contribution of the study.

2. The lack of context of the data retrieved from different platforms makes the experiment procedure suspicious and confusing. For example, from the data collected from Stack Overflow, are these singular-type questions or multi-hop questions? What were the subjects of the transcribed videos and answers from Youtube and Yahoo? These details are important.

3. It was mentioned that the use of Roget’s thesaurus is to map words to related categories for a thematic-style analysis. But why are other, more common forms of thematic analysis not explored such as LDA, BERTopic, or Contextual Topic Models? These might even give better results on the differences between theme use as evidenced by the results presented in their corresponding papers (especially CTM).

4. What I understand from the classification phase is that the author/s uses TF-IDF as features for the models. Why not combine the other linguistic features used? There is no exploration or ablation from this particular experiment.

---

### Official Review · Reviewer_k3D5 · 2023-10-31

**Soundness:** 3 good
**Presentation:** 2 fair
**Contribution:** 2 fair
**Rating:** 3
**Confidence:** 4

**Summary:**

The authors construct a new dataset containing human-generated text and ChatGPT-generated text, and compare them though conceptual, contextual, or lexicographic factors with rich experiments. The results show that syntactic and contextual differences are insignificant but conceptual and lexicographic differences appear to be more significant.

**Strengths:**

1. This paper introduces novel parallel datasets of human and ChatGPT texts on various domains and tasks, which can be useful for future research on text generation and evaluation.

2. This paper employs a comprehensive and diverse set of evaluation metrics, conducting rich experiments to analyze the similarity and differences between human and ChatGPT texts. The experimental results are intuitive.

3. This paper is easy to follow.

**Weaknesses:**

1. My main concern is that the paper does not provide in-depth and significant features or differences between human-generated text and ChatGPT-generated text. This leads readers to find it difficult to obtain sufficient information from the paper.

2. Some findings are not rigorous. For example, although the author asserts the differences on the words based on Figure 6, using only two examples and a lesser-used metric "WORD NEIGHBORHOODS" is not sufficient to adequately demonstrate this point.

3. The experiment is done on various metrics, unveiling whether human and ChatGPT texts are significantly different on the corresponding factors. However, the authors do not delve deeper to explain why these kinds of similarity or difference happen.

4. In section 4.1, the author classify the documents with machine learning models. The usage of BERT is not explained clearly enough. Does BERT embeddings are simply used for classification without other machine learning models? If that, which kind of classifier is used?

**Questions:**

Please refer to the weaknesses.

---

### Official Review · Reviewer_UaJh · 2023-11-02

**Soundness:** 2 fair
**Presentation:** 3 good
**Contribution:** 1 poor
**Rating:** 5
**Confidence:** 2

**Summary:**

Authors study similarities/ differences between human-generated and GPT-generated text. They use samples from stack overflow, yahoo answers and youtube, and compare POS tag distribution, BLEU score, stopword distribution, embedding-based similarity score, ML-classifiers, neighbor words distribution and categories in Roget's thesaurus between human- and GPT- generated texts. They find that both human- and GPT- generated texts

1) have high embedding-based similarity and similar POS-tag and stopwords distributions, implying insignificant syntactic and contextual differences.

2) are easily distinguishable by ML-classifiers, implying high lerxicographic differences.

3) have distinct word-usage patterns, as indicated by Roget's categories and neighborhood words distributions.

**Strengths:**

A systematic study to analyze the similarity between human-generated and GPT-generated text.

**Weaknesses:**

- The motivation of this study is unclear. Even texts generated by two humans (depending on their geography/ dialects/ etc.) would have some differences and similarities. Having a clearly defined objective can help better understand both the design of experiments and the significance of results.
For instance, if the goal was to evaluate whether an LLM can answer like human for stack overflow questions (or any specific domain), performing these analyses on fine-tuned LLM (or few-shot/ incontext examples) can give more useful insights.

**Questions:**

- What is the significance of the experiments and analyses conducted in this study? Could this work be potentially a better fit for linguistics-related conferences?

- Did you try comparing two different LLMs (e.g., chat-GPT vs GPT-4) with the same prompt (not CGPT3 vs CGPT3-C)? Are they similar or show distinct patterns? Or what if we compare top 2 human-written answers for stack overflow questions, do they show similar distributions?

---

### Official Review · Reviewer_3Tnm · 2023-11-05

**Soundness:** 1 poor
**Presentation:** 1 poor
**Contribution:** 1 poor
**Rating:** 1
**Confidence:** 5

**Summary:**

This paper compares human-generated content to ChatGPT-3.5, ChatGPT-4, and Davinci-3 model generated content. It uses questions from StackOverflow and Yahoo Answers as well as YouTube dataset to generate the answers. The paper utilizes Roget’s thesaurus and some lexical, statistical and sematic methods to uncover thematic similarities and differences between the human corpora and GPT corpora. The paper finds some surface level differences between the two but suggests that at a semantic level, there's a lot of similarity. The paper also uses some ML models and is able to distinguish between the two generations with decent F1 scores.

**Strengths:**

- the main contribution seems to be the creation of the dataset corpora for running comparative evaluations
- the paper uses multiple axes of examination at a surface level (lexical, syntactic, semantic, using ML classifies, using Roget's thesaurus) and shows the experiments on comparisons on the humans vs ChatGPT output.

**Weaknesses:**

- the paper needs significant scientific rigor in the analysis to highlight the similarities and differences in how humans vs SoTA LLMs generate text. There a lot of similar work out there (even the two comparative studies the authors point out have similar analysis and rigorous insights on experiments and results)
- It is not clear if Fig 1 uses single or combined human and chatgpt corpus categories. The figure does not add value to the main content of the paper (could perhaps go in appendix).
- It is interesting to see that YA-D3 vs YA-H have the least in common in terms of semantic textual similarity (Fig 4) - this could have been an interesting topic to investigate further, do some human evaluations, etc.
- Fig 5 uses T-SNE projections; some experiments suggest that using UMAP visualization might show better projections - also, there are many advanced methods (such as Deep Aligned Clustering, for example) that can be used to further cluster the semantic embeddings to get a better understanding of topics and overlap between the two datasets.
- Can’t draw much insights from Fig 6 and it needs more explanation (what are the colors here?)

**Questions:**

(please see weaknesses above)
- the conclusions drawn form the paper are very generic, coming up with more specific guidelines and measures for identifying similarities and differences would be a nice contribution.
- The paper does not discuss the current SoTA in AI generated content detection which is a very important topic of study (also from AI regulations perspective)

---

### Meta-Review · Area_Chair_7JCy · 2023-12-07

**Metareview:**

This paper presents analysis of a few different methods on identifying human-generated vs model-generated text. Authors show that using simple word counting based methods can distinguish between the two texts as the LLM generated responses contain more common words and human generated responses contain more unique words.

This paper lacks scientific rigor, it doesn't reference any previous works or includes those numbers in baselines. The approach seems like a mix-n-match of available tools. And the paper draws incorrect conclusions. Differentiating between human and LLM generated content is an extremely difficult task, it cannot be solved by word counting. I agree with the criticisms by the reviewers.

**Justification For Why Not Higher Score:**

This is not a scientific study.

**Justification For Why Not Lower Score:**

n/a

---

### Decision · Program_Chairs · 2024-01-16

Reject